# Relationship between Quality of Nursing Work Life and Uniformed Nurses’ Attitudes and Practices Related to COVID-19 in the Philippines: A Cross-Sectional Study

**DOI:** 10.3390/ijerph18199953

**Published:** 2021-09-22

**Authors:** Juneffer Villamen Navales, Amadou Wurry Jallow, Chien Yu Lai, Chieh Yu Liu, Shu Wen Chen

**Affiliations:** 1School of Nursing, National Taipei University of Nursing and Health Sciences, Taipei City 112303, Taiwan; jvnavales76@yahoo.com (J.V.N.); amswurryjallow@gmail.com (A.W.J.); chienlai@ntunhs.edu.tw (C.Y.L.); chiehyu@ntunhs.edu.tw (C.Y.L.); 2Philippine Coast Guard Manila, Metro Manila, Manila 1018, Philippines; 3Research Center for Healthcare Industry Innovation, National Taipei University of Nursing and Health Sciences, Taipei City 112303, Taiwan

**Keywords:** quality of nursing work life (QNWL), COVID-19, uniformed nurses, attitude, practice

## Abstract

(1) Background: Coronavirus disease 2019 (COVID-19) has spread rapidly worldwide. Uniformed nurses have played a critical role during the COVID-19 pandemic in the Philippines; however, uptake of literature is limited. This study assessed the relationship between quality of nursing work life (QNWL) and nurses’ attitudes and practices during the COVID-19 pandemic. (2) Methods: A descriptive cross-sectional design was used. Participants were recruited from four government hospitals in the Manila metropolitan area of the Philippines. Participants completed three questionnaires in an online survey: a demographic questionnaire, a QNWL questionnaire, and the attitude and practices toward COVID-19 questionnaire. Descriptive statistics, an independent *t*-test, a one-way analysis of variance, the Pearson correlation coefficient, and hierarchical linear regression were applied for data analysis. (3) Results: The mean age of the participants was 29 years. Most of the participants were single women who were not certified in their specialties. A total of QNWL scores were high, indicating that the participants displayed favorable attitudes and practices in relation to COVID-19. A statistically significant relationship was observed between QNWL, specialty certification, and practices related to COVID-19. Practices related to COVID-19 were a significant predictor of QNWL and one of its subscales, work design. (4) Conclusion: Young adult uniformed nurses in the Philippines have assumed numerous responsibilities during the COVID-19 pandemic. Providing these frontline nurses with comprehensive specialized education and training is crucial.

## 1. Introduction

Coronavirus disease 2019 (COVID-19) continues to spread worldwide at an unprecedented rate. According to a COVID-19 dashboard report by the Center for Systems Science and Engineering at John Hopkins University, as of 21 July 2021, the number of global COVID-19 cases had reached 186,800,826, and the number of deaths had reached 4,031,654 [1]. COVID-19 has substantially affected the global economy, finance, and the trade of goods and services [2]. Most countries have been affected by the COVID-19 pandemic, which has resulted in the closure of educational, commercial, sports, and spiritual organizations. In addition, the business transportation industry, including international and domestic flights, has been considerably demobilized [3].

Frontline health care workers (HCW) have not been spared from this period of high morbidity and mortality [4,5,6]. A meta-analysis study that investigated the clinical outcomes and risk factors of SARS-CoV-2 infection in HWC indicated that the prevalence of hospitalization and mortality among infected HCW during the first six months of the pandemic were 15.1% and 1.5%, respectively [7]. Infection among HCW was attributed to a lack of personal protective equipment (PPE) during the first stage of the crisis, prolonged exposure to high numbers of infected patients, a shortage of PPE throughout the intensification of the outbreak, and a lack of knowledge and training among health care providers regarding infection prevention control [8,9]. Nurses were the most affected personnel working in hospitals or nonemergency wards [10]. HCW were also at relatively high risk for COVID-19 because of their workplace setting, profession, contacts, and testing [7].

Quality of nursing work life (QNWL) refers to the job experience of licensed nurses and whether individual and organizational objectives are fulfilled [11,12,13]. QNWL assessment comprises four dimensions: work life and home life, work design, work context, and work world [12]. Work life and home life refer to the interface between nurses’ work and home life. Work design refers to the nature and contents of nursing work. Work context refers to the environment in which nurses work and its impact on both nurse and patient systems. Work world refers to the effects of broad societal influences and changes on nursing practice [12]. The concept of QNWL has been applied in many countries to improve the quality of nursing life, including Saudi Arabia, Jordan, India, Bangladesh, Singapore, and China [14,15,16,17,18]. Several factors were related to QNWL. In Indonesia, the individual factors are related with nurses’ QNWL, including the older nurses, female, bachelor graduated with more dependents, more children, have job positions with longer work experience, and the highly motivated tend to have better QNWL [19]. In Arabia, the majority of nurses were aged between 36 and 46 years, were married, and were taking care of children younger than four years of age or elderly parents, indicating moderate QNWL [20]. In Jordan, young nurses working in emergency rooms exhibited moderate QNWL [16]. In India, nurses working in public hospitals had higher QNWL than did those working in private hospitals [18]. Nurses working in Bangladesh had moderate QNWL, and monthly income was the best predictor of QNWL [14].

During the COVID-19 pandemic, nurses’ attitudes and practices have strongly affected the quality of their work life. One systematic review assessed health professionals’ knowledge, attitudes, and practices (KAP) related to COVID-19 [4]. Approximately 80% of health professionals demonstrated adequate knowledge of COVID-19 symptoms (79%) and its transmission (82%), and they avoided crowded environments to prevent COVID-19 infection (89%) [4]. Another systematic review and meta-analysis were conducted in Ethiopia and showed an adequate level of knowledge and positive attitudes but poor overall practices [5]. Similarly, one cross-sectional KAP survey indicated that HCW possessed adequate knowledge and a positive attitude but exhibited poor practices concerning COVID-19 in Uganda [6].

A total of 1,403,025 cases and 25,921 deaths in the Philippines were recorded by the Center for Systems Science and Engineering at John Hopkins University on 21 July 2021 [1]. Uniformed nurses work for the military, police, and coast guard to address health needs and provide high-quality care [21]. Uniformed nurses have encountered myriad challenges during the COVID-19 pandemic. Although numerous quantitative studies have focused on QNWL [15,16,19] and addressed KAP related to COVID-19 [4,5,6], few studies have identified the relationship between QNWL and nurses’ attitudes and practices related to COVID-19, especially those of uniformed nurses. To address this gap, this study assessed the relationship between QNWL and uniformed nurses’ attitudes and practices related to COVID-19.

## 2. Methods

A quantitative, descriptive, cross-sectional design was used to determine the relationship between QNWL and uniformed nurses’ attitudes and practices related to COVID-19 in the Philippines.

### 2.1. Participants and Eligibility Criteria

This study recruited natural-born Filipino citizens aged 18 years and older who held a bachelor’s degree, were employed as uniformed or military nursing staff, and agreed to participate. Temporarily employed non-uniformed nurses, head nurses, and nurse managers were excluded.

### 2.2. Study Setting

This study was conducted in four government hospitals, namely Victoriano Luna Medical Center, Philippine National Police General Hospital, Philippine Coast Guard Medical Service, and Philippine Airforce General Hospital, which are located in the Manila metropolitan area of the Philippines.

### 2.3. Sampling and Sampling Size

The sample size was calculated using G*Power (version 3.1). According to the previous literature [21,22], the medium effect size = 0.15, a total of 135 samples would be needed to provide α = 0.05, and power = 0.80. The anticipated ineligibility was 10%, and thus, a sample size of 149 was required. A total of 147 nurses participated in this research.

### 2.4. Instrument

The questionnaires were divided into two parts: Brooks’ QNWL questionnaire and the attitudes and practices related to COVID-19 questionnaire.

### 2.5. QNWL Questionnaire

The QNWL questionnaire is a self-reported questionnaire consisting of two parts: demographic data and Brooks’ QNWL survey. The QNWL questionnaire consists of 42 items in four subscales. Each item is measured using a 6-point Likert-type scale with 1 indicating strongly disagree, 2 indicating moderately disagree, 3 indicating disagree, 4 indicating agree, 5 indicating moderately agree, and 6 indicating strongly agree. The total score ranges from 42 to 252 points, and results are divided into low, moderate, and high scores. A higher score indicates higher QNWL. The subscales consisted of 7 items for work life and home life, 10 items for work design, 20 items for work context, and 5 items for work world.

Studies have validated the scale’s internal consistency reliability, finding a Cronbach’s α of 0.89 [11]. A study of 53 registered nurses over a 14-day interval indicated high test-retest reliability for the total QNWL score (*r* = 0.90, *p* < 0.001) [11]. In another study conducted in Saudi Arabia, the internal consistency reliability of the total QNWL scores was 0.89 [23]. One study indicated validity with a significant positive Pearson correlation (*r* = 0.72, *p* < 0.01) between QNWL and the practice environment scale [24].

### 2.6. Attitudes and Practices Related to COVID-19 Questionnaire

The attitudes and practices related to COVID-19 questionnaire was adapted from the KAP related to COVID-19 questionnaires [25]. Attitudes toward COVID-19 was measured through two questions to understand whether participants agreed with the efficacy of the control of COVID-19 and whether they had confidence in effectively managing COVID-19. Practices were assessed by using two representative behaviors: going to crowded places and wearing a mask when going out [25]. The reliability of the questionnaire was tested in a pilot study on 40 students conducted in Pakistan, which demonstrated a Cronbach’s α of 0.79 [26].

A self-reported questionnaire was used to measure attitudes and practices related to COVID-19. The questionnaire was adopted from published articles and modified to suit this study [27]. Attitudes toward COVID-19 were measured using two questions. The first question, A.1, was “Do you agree that COVID-19 will be successfully controlled?” The answers were “I don’t know,” “disagree,” and “agree,” which were represented as 0, 1, and 2 points, respectively. The second question, A.2, was “Do you have confidence that the Philippines can win the battle against COVID-19?” The answers were “no” and “yes,” which were represented by 1 and 2 points, respectively. Practices related to COVID-19 were measured using two questions. The first question, P.1, was “Have you recently gone to any crowded places?” The answers were “yes” and “no,” which were represented by 1 and 2 points, respectively. The second question, P.2, was “Have you recently worn a mask when leaving home?” The answers were “no” and “yes,” which represented 1 and 2 points, respectively. The total attitude and practice scores were between 0 and 4 points. A score of ≤2 points indicated a negative attitude, and a score of >2 points indicated favorable practices.

### 2.7. Data Collection

Data were collected through an online survey conducted between July 2020 and January 2021 during the COVID-19 pandemic. The researcher sent an email and a link to the survey to a chief nurse at each hospital. After the chief nurses received the email, they forwarded the information to the other three head nurses of the hospitals through an online chat group. The participants were informed of the survey through personal emails and accessed the survey on a Google website. The participants read the informed consent and clicked “next” if they agreed to participate in the study (Figure 1).

### 2.8. Data Analysis

The data were managed and analyzed using SPSS (version 26.0, IBM, Taipei, Taiwan).

Descriptive statistics was employed to identify the frequencies and percentages of the demographic characteristics. Means with standard deviations (SDs) were used to describe the continuous variables, namely QNWL and attitudes and practices related to COVID-19. An independent *t*-test, the Pearson correlation coefficient, and a one-way analysis of variance (ANOVA) were used to determine the relationships among the demographic characteristics, QNWL, and attitudes and practices related to COVID-19. To predict the factors affecting QNWL, multiple hierarchical linear regression was employed.

### 2.9. Ethical Considerations

Ethical approval was obtained from the Institutional Review Board of the Philippine Coast Guard and National Taipei University of Nursing and Health Sciences. The participants were provided with online informed consent prior to the study. Confidentiality was assured to protect the participants’ right to privacy.

## 3. Results

### 3.1. Demographic Information of Participants

More than half the participants were aged between 20 and 30 years; were women (53.70%; *n* = 79); were single (57.1%; *n* = 84); had no children (50.30%; *n* = 74); were not taking care of spouses, partners, or elderly parents (85%; *n* = 125); belonged to the major ethnic groups (55.80%; *n* = 82); and worked rotating shifts (89.10%; *n* = 131). The majority had mandatory rotating shifts (92.50%; *n* = 136) and received no additional compensation for rotating shifts (88.40%; *n* = 130). Almost half of the participants were working in an in-patient department (48.30%; *n* = 71). Less than 10% of participants (96.60%; *n* = 142) were certified in specialty areas, and hence, almost all received no additional compensation for being certified (97.30%; *n* = 143; Table 1 and Figure 2).

### 3.2. Descriptive Analysis of Characteristics for QNWL, Attitudes, and Practices Related to COVID-19

QNWL is presented as means with SDs. The total score for QNWL ranged from 42 to 252 points. The range of 42 to 112 points represented low QNWL, 113 to 182 points represented moderate QNWL, and 183 to 252 points represented high QNWL. In the present study, the average score was 185.56 points (SD = 22.52), representing high QNWL. The QNWL subscales indicated moderate level of work life and home life (29.4 points; SD = 3.35) and work design (43.66 points; SD = 4.12), high quality of work context (91.65 points; SD = 14.61), and low quality of work world (20.85 points; SD = 3.50; Figure 3).

The mean score for attitudes was 3.14 points (SD = 0.93), and that for practices was 2.98 points (SD = 0.70), indicating favorable attitudes and practices, respectively. Most of the participants (72.10%; *n* = 106) answered “I don’t know” for A.1 (Do you agree that COVID-19 will be successfully controlled?), and more than half (55.1%; *n* = 81) answered “yes” for A.2 (Do you have confidence that the Philippines can win the battle against COVID-19?). For practices, more than half (62.60%; *n* = 92) indicated that they had not gone to crowded places (P.1) and had worn masks when leaving home (P.2) (60.50%; *n* = 89).

### 3.3. Demographic Characteristics and its Association with QNWL, Attitudes, and Practices Related to COVID-19

The means and SDs of age; gender; marital status; number of children; care for spouses, partners, or elderly parents; ethnicity; involvement in rotating shifts; willingness to participate in rotating shifts; additional compensation for rotating shifts; unit type; specialty certifications; and additional compensation for certification were calculated. The *t*-test and one-way ANOVA for the demographic characteristics revealed a significant relationship between QNWL score and specialty certification (*t* = 2.64, *p* = 0.04 *) (Table 2). The results indicated no significant relationship between the demographic variables and attitudes toward COVID-19. However, five demographic variables were more likely to be associated with practices related to COVID-19: age (*p* = 0.01 *); care for spouses, partners, or elderly parents (*p* < 0.00 **); ethnicity (*p* < 0.00 ***); involvement in rotating shifts (*p* = 0.03 *); and unit type (*p* < 0.00 ***) (Table 2).

### 3.4. Factors Predicting QNWL

A hierarchical multiple linear regression was used to identify the most effective predictors for QNWL. The total QNWL score was assigned as the dependent variable. The data were represented using standard error, standardized betas (β), *t*-values (*t*), and *p*-values (*p*). The unadjusted model was model 1, whereas models 2 and 3 were adjusted models. The variables for demographic characteristics were impute into model 1, total attitude score was added into model 2, and total practice score was added into model 3. However, no statistically significant result was observed in any of the models.

The hierarchical multiple linear regression was performed with three variables (gender, age, and ethnicity) for demographic characteristics in model 1 (unadjusted model), model 2 (model 1 adjusted with total attitude score), and model 3 (model 2 adjusted with total practice score). Gender, age, and ethnicity were not statistically significant in models 1 and 2 (Table 3). However, in model 3, a statistically significant difference was observed in total practice score, with standardized coefficient of β = 0.17 and *p* = 0.04. Practices related to COVID-19 were the most robust predictor of QNWL. Attitude in model 2 was not significant even after adjustment for gender, age, and ethnicity (Table 3).

### 3.5. Predicting Factors of QNWL Subscales

Another hierarchical multiple linear regression was employed to identify the predicting factors of the QNWL subscales. The same steps were applied to analyze model 1 (unadjusted, gender, age and ethnicity), model 2 (model 1 adjusted with total attitude score), and model 3 (model 2 adjusted with total practice score). Only work design (*p* < 0.00 ***) was statistically significant (Table 4).

## 4. Discussion

Uniformed nurses have encountered myriad challenges during the COVID-19 pandemic. Working in a stressful environment during stressful time has a detrimental effect on health care providers’ quality of life and professional activities [28]. This study assessed the relationship between QNWL and nurses’ attitudes and practices related to COVID-19 during the pandemic. Risk perception refers to individuals’ evaluation of potential hazards [29] and their estimation of the probability of encountering danger and its consequences [30]. Individuals perceive risk and respond by interacting with others and through cultural adherence [20,31,32,33]. Perceptions of health risk are influenced by individuals’ beliefs and knowledge regarding health [34]. The results of this study indicate that uniformed nurses had positive attitudes and displayed effective practices in relation to COVID-19 and had high QNWL.

### 4.1. Demographic Characteristics and QNWL

Total QNWL scores were high. More than half of uniformed nurses were young adults and single. The scores indicate that they are committed to their organizations and satisfied with their working environments. However, the results are inconsistent with those of other studies [15]. In a study in Saudi Arabia, most nurses were aged between 36 and 46 years, were married, and were taking care of children younger than four years of age or elderly parents, indicating moderate QNWL [20]. In addition, young nurses working in emergency rooms in Jordan exhibited moderate QNWL [16]. Our study yielded higher QNWL scores because the uniformed nurses reported that they received adequate supervision from their managers and supervisors, were friendly with their coworkers, and cooperated in the work environment. Certain features of hospitals also influence QNWL. In India, nurses working in public hospitals had higher QNWL than did those working in private hospitals [18]. Because the Philippine government increased salaries for the military, police, and coast guard, with such increases applying to the health care professionals in these fields, uniformed nurses in hospitals under these branches have higher salaries than do those working in general hospitals. In our study, almost 43% of nurses were married, and merely nine nurses were taking care of elderly parents. Of 46.9% (*n* = 63) of nurses, 7 nurses were not taking care of children, 55 nurses were not taking care of 1–3 children, and just 1 nurse was taking care of more than 3 children. For the majority of married nurses, a satisfying salary might explain the high QNWL scores in this study. Moreover, one study indicated that nurses working in Bangladesh had moderate QNWL and that monthly income was the best predictor of QNWL [14]. To provide family income, frontline nurses were willing to work during the COVID-19 pandemic in China [35]. One study suggested that motivation fosters nurses to perform their duties and receive rewards and promotions from their superiors, which can improve QNWL [19]. One study conducted in China revealed that most frontline nurses (96.8%) were willing to work during the COVID-19 pandemic [35] and identified five predictors of their willingness to work: monthly family income, average working hours per shift, belief in their colleagues’ preparedness, belief in their hospitals’ preparedness, and level of depression [35]. Despite the perceived risk of COVID-19, HCW remain willing to work during the COVID-19 pandemic because of their professional commitment, satisfaction with their income, and ability to care for patients.

Specialty certification was the only factor associated with QNWL in our study. In Saudi Arabia, specialty units were a significant factor contributing to higher QNWL scores [15]. Fear may affect the COVID-19 infection rates among frontline HCWs. HCWs have been found to fear infection, failing to provide adequate care for patients with limited resources, bringing the virus back to their homes and infecting family and friends, and stigmatization [36]. Well-designed and rigorous training programs can improve QNWL [16,37]. Thorough training in specialty areas strengthens nurses’ knowledge and skills to respond to COVID-19 and provide high-quality care [38].

### 4.2. Demographic Characteristics and Attitudes and Practices Related to COVID-19

Uniformed nurses exhibited favorable attitudes and practices related to COVID-19. The results are consistent with a systematic review that revealed positive attitudes and appropriate practices among physicians, HCWs, and the general population regarding COVID-19 in several countries [39]. In the Philippines, the government implemented strict public health protocols, such as the requirement to wear face masks and face shields outside the home, social distancing, and frequent hand washing during the COVID-19 pandemic [40]. Despites this finding, attitudes toward COVID-19 were not statistically significant.

Several factors were associated with practices related to COVID-19. Age was significantly associated with practices related to COVID-19. A study conducted in Saudi Arabia revealed that the majority of young nursing students, with a mean age of 23 years, avoided going to crowded places but did not wash their hands frequently [41]. A similar study in northern Nigeria revealed that the majority of HCWs, with a mean age of 28.58 years, believed that implementing strict health care measures in response to COVID-19 was crucial [42]. Because most of the uniformed nurses in this study were not taking care of spouses, partners, or elderly parents, they had sufficient time to dedicate themselves to their work. Several studies have demonstrated that nurses had trouble balancing their commitments to family and work and that heavy workloads tend to induce them to find other jobs [17,23,43]. One study conducted in Singapore revealed that nurses who spent more time at work had less time for their private lives and lower job satisfaction. Strategies for improving the QNWL of those caring for relatives at home may require support from family, coworkers, and administrators for nurses to manage their stress [17]. Involvement in rotating shifts was associated with COVID-19 practices in this study, which is consistent with the results of the study conducted in Saudi Arabia [15]. In Greece, shift work negatively affected sleep quality and the quality of work life for health care professionals [44]. One study reported that shift work involving physicians and nurses working 12 h every three days was essential for maintaining the integrity of the workforce and reducing the probability of team failure and infection during the COVID-19 pandemic [45].

### 4.3. Predictors of QNWL

In this study, practices related to COVID-19 were the strongest significant predictor of QNWL. Appropriate practices related to COVID-19 were more likely to be associated with QNWL. Because the COVID-19 virus is 5 to 10 μm, which is large enough to fall onto surfaces, individuals may be infected by touching surfaces with the virus and then touching their faces [46]. This might be the motivation for the new rules established by the Philippine government that require wearing a face mask and face shield in public and the stricter lockdowns to prevent hospitals from being overwhelmed by patients. Studies are investigating distancing of 2 m and the wearing of masks, which might prevent infection among nurses. In a study conducted in Lebanon, most of the nurses wore face masks for up to 3 h before disposal, more than half had received infection prevention and control training in Saudi Arabia, and the majority reported implementing strict preventive practices [39,47]. A study in Thailand suggested that mindfulness should be practiced in response to the fear and anxiety associated with wearing face masks, PPE, and hand washing. Another study indicated that social distancing and simply avoiding crowded places help prevent the spread of infection [48].

In this study, work design was a significant predictor for QNWL, which suggested that nurses were satisfied with their job, received sufficient assistance, had ample time to perform their duties, and were confident in their ability to provide high-quality care to their patients. However, in a study conducted in Bangladesh, QNWL was predicted by income, work environment, organizational commitment, and job stress [14]. By contrast, increased workloads and nurse shortages were the dominant factors in the domain of work design in a study in Jordan [16].

### 4.4. Limitations

This study has several limitations. First, the ongoing COVID-19 pandemic limited the collection of data. The participants were affected by lockdowns, and some were infected with COVID-19. Thus, an online survey was used, and the relatively small sample size may limit the generalizability of the findings. In addition, online surveys (emails and group chat messengers) may result in bias because certain individuals may have experienced network connection problems that prevented participation. Demographic information of the number of COVID-19 patients, the ratio of nurses and patients, and the features of hospitals may be related to QNWL, which are needed to be further examine in future studies. Vaccination is an effective strategy to combat COVID-19. However, vaccination was not included as a factor in the attitudes and practices questionnaire. Subsequent studies should address the relationship between vaccination and QNWL. Despites these limitations, this study can encourage researchers to conduct conscientious research on education and training courses as interventions to improve QNWL.

## 5. Conclusions

This study broadly assessed the relationship between QNWL and nurses’ attitudes and practices related to COVID-19. The majority of uniformed nurses in this study prioritized health and safety. Although they had high QNWL and displayed positive attitudes and effective practices in relation to COVID-19, they were not certified in their specialties. During the COVID pandemic, QNWL can be improved by providing comprehensive and continuous specialized education and training and strengthening nurses’ cognitive skills and abilities to match global standards.

## Figures and Tables

**Figure 1 ijerph-18-09953-f001:**
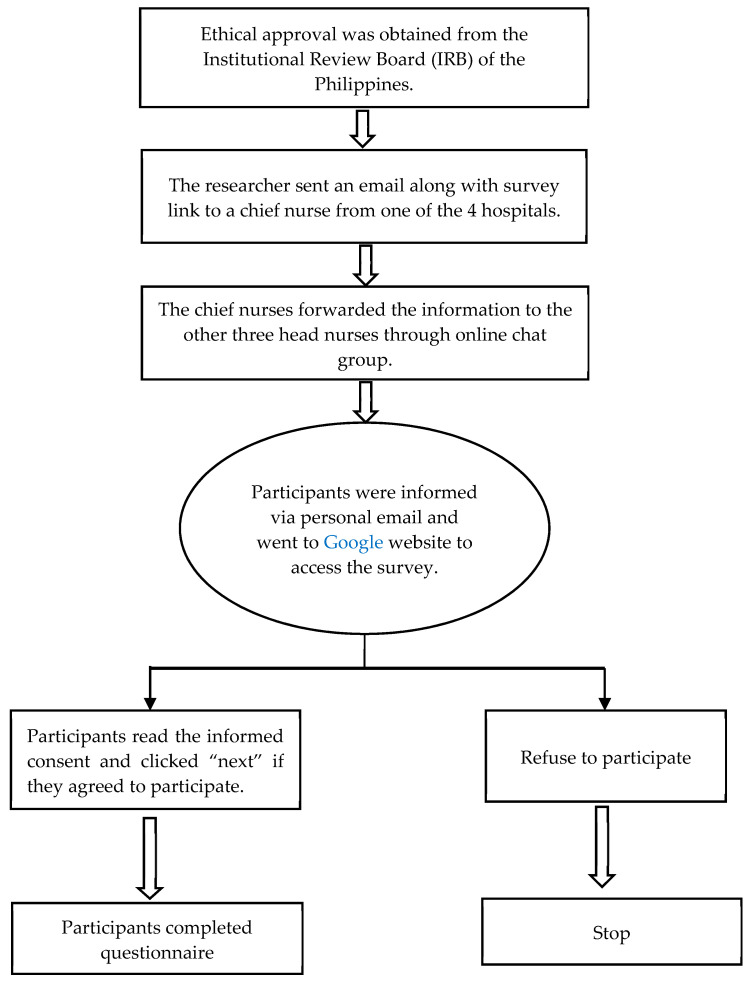
Flow diagram of data collection.

**Figure 2 ijerph-18-09953-f002:**
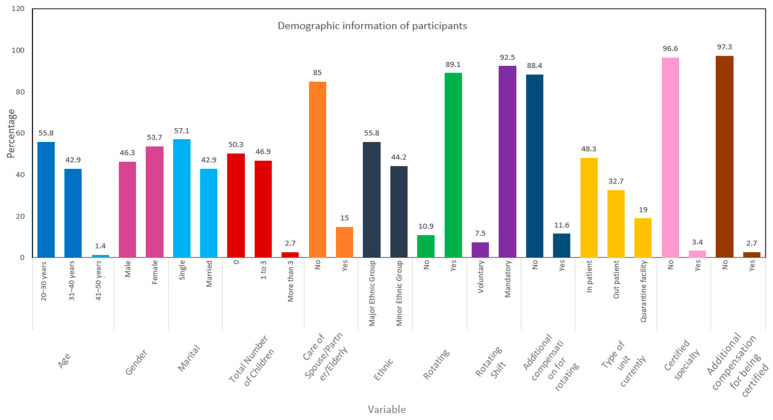
Comparison of demographic characteristics of participants (*n* =147).

**Figure 3 ijerph-18-09953-f003:**
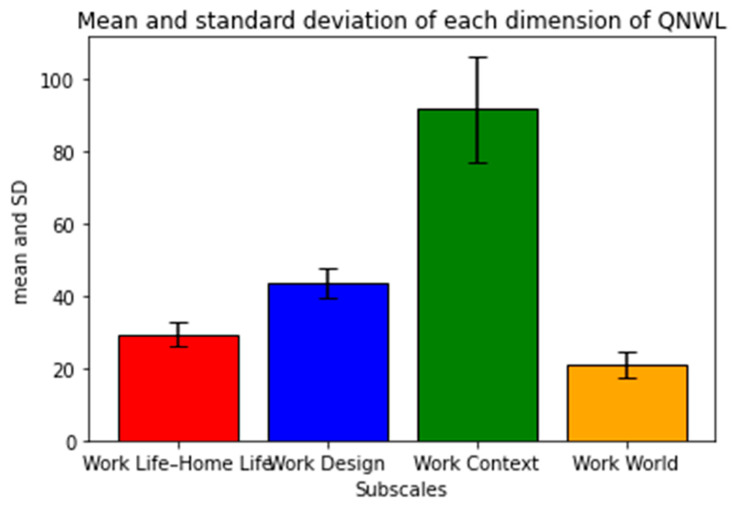
Mean and standard deviation of each dimension of QNWL (*n* = 147).

**Table 1 ijerph-18-09953-t001:** Demographic characteristics of participants (*n* =147).

Variable	*n*	%
Age		
20–30 years	82	55.80
31–40 years	63	42.90
41–50 years	2	1.40
Gender		
Male	68	46.30
Female	79	53.70
Marital Status		
Single	84	57.10
Married	63	42.90
Total Number of Children		
0	74	50.30
1 to 3	69	46.90
More than 3	4	2.70
Care of Spouse/Partner/Elderly Parents		
No	125	85
Yes	22	15
Ethnic Group		
Major Ethnic Group	82	55.80
Minor Ethnic Group	65	44.20
Rotating Shift		
No	16	10.90
Yes	131	89.10
Rotating Shift Willingness		
Voluntary	11	7.50
Mandatory	136	92.50
Additional compensation for rotating shifts.		
No	130	88.40
Yes	17	11.60
Type of unit currently working.		
In patient	71	48.30
Outpatient	48	32.70
Quarantine facility	28	19
Certified specialty area.		
No	142	96.60
Yes	5	3.40
Additional compensation for being certified.		
No	143	97.30
Yes	4	2.70

**Table 2 ijerph-18-09953-t002:** Univariate analysis for QNWL, attitudes, and practices related to COVID-19.

Variables	QNWL			Attitudes			Practices		
	Mean (SD)	t/F/r	*p*-Value	Mean (SD)	t/*F*/*r*	*p*-Value	Mean (SD)	t/F/r	*p*-Value
Age (Continuous Data)	29.15 (4.02)	0.10	0.21	29.15 (4.02)	−0.05	0.58	29.15 (4.02)	−0.21	0.01 *
Care of Spouse/Partner/Elderly Parents		0.12	0.91		1.14	0.26		3.29	0.00 **
Ethnicity		0.07	0.95		0.38	0.71		−4.04	0.00 ***
Rotating Shift		−1.04	0.31		−0.91	0.38		−2.45	0.03 *
Unit		2.90	0.06		2.27	0.11		17.22	0.00 ***
Certified specialty area		2.64	0.04 *		0.17	0.87		−1.09	0.33

* *p* < 0.05, ** *p* < 0.01, *** *p* < 0.001.

**Table 3 ijerph-18-09953-t003:** Predicting Factors of QNWL (*n* = 147).

Model	Variable	SE	Beta	t	*p*-Value
1	(Constant)	14.50		11.76	0
	Gender	3.79	0.10	1.15	0.25
	Age	0.47	0.09	1.03	0.31
	Ethnicity	3.75	−0.02	−0.18	0.86
2	(Constant)	15.99		10.30	0
	Gender	3.84	0.11	1.26	0.21
	Age	0.47	0.09	1.04	0.30
	Ethnicity	3.76	−0.02	−0.21	0.83
	Total Attitudes Scores	2.04	0.07	0.85	0.40
3	(Constant)	18.60		7.65	0
	Gender	3.79	0.12	1.44	0.15
	Age	0.48	0.13	1.54	0.13
	Ethnicity	3.94	−0.09	−0.98	0.33
	Total Attitudes Scores	2.01	0.07	0.87	0.38
	Total Practices Scores	2.88	0.20	2.27	0.03 *

* *p* < 0.05.

**Table 4 ijerph-18-09953-t004:** Predicting factors for subscales, QNWL.

		Work Life–Home Life				Work Context				Work World				Work Design			
Model	Variable	SE	*Beta*	*t*	*p*-Value	SE	*Beta*	*t*	*p*-Value	SE	*Beta*	*t*	*p*-Value	SE	*Beta*	*t*	*p*-Value
1	(Constant)	2.17		12.39	0	9.40		8.71	0	2.22		7.47	0	2.67		16.88	0
	Gender	0.57	0.02	0.26	0.80	2.46	0.10	1.15	0.25	0.58	0.13	1.51	0.13	0.70	0.06	0.71	0.48
	Age	0.07	0.10	1.15	0.25	0.31	0.09	1.05	0.30	0.07	0.16	1.91	0.06	0.09	−0.05	−0.64	0.53
	Ethnicity	0.56	0.01	0.13	0.90	2.43	−0.02	−0.25	0.81	0.57	−0.02	−0.18	0.86	0.69	0.00	−0.05	0.97
2	(Constant)	2.38		10.64	0	10.36		7.50	0	2.45		6.45	0	2.95		15.55	0
	Gender	0.57	0.04	0.48	0.63	2.49	0.11	1.28	0.20	0.59	0.14	1.60	0.11	0.71	0.05	0.61	0.55
	Age	0.07	0.10	1.19	0.24	0.31	0.09	1.07	0.29	0.07	0.16	1.92	0.06	0.09	−0.06	−0.65	0.52
	Ethnicity	0.56	0.01	0.06	0.95	2.43	−0.02	−0.29	0.78	0.58	−0.02	−0.21	0.83	0.69	0.00	−0.02	0.99
Total Attitudes scores																	
		0.30	0.13	1.55	0.12	1.32	0.08	0.96	0.34	0.31	0.06	0.75	0.45	0.38	−0.05	−0.64	0.52
3	(Constant)	2.78		8.11	0	12.15		5.52	0	2.90		5.11	0	3.25		11.63	0
	Gender	0.57	0.05	0.62	0.53	2.48	0.12	1.41	0.16	0.59	0.14	1.64	0.10	0.66	0.08	1.00	0.32
	Age	0.07	0.14	1.58	0.12	0.31	0.12	1.41	0.16	0.07	0.17	2.01	0.05	0.08	0.03	0.36	0.72
	Ethnicity	0.59	−0.05	−0.58	0.56	2.58	−0.07	−0.83	0.41	0.61	−0.04	−0.42	0.68	0.69	−0.14	−1.61	0.11
	Total Attitudes scores																
		0.3	0.13	1.57	0.12	1.31	0.08	0.97	0.33	0.31	0.06	0.75	0.45	0.35	−0.05	−0.66	0.51
	Total Practices scores																
		0.43	0.17	1.87	0.06	1.88	0.15	1.65	0.10	0.45	0.06	0.64	0.52	0.50	0.40	4.66	0.00 ***

*** *p* < 0.001.

## Data Availability

Please refer to suggested Data Availability Statements in section “MDPI Research Data Policies” at https://www.mdpi.com/ethics.

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
