# Peer review of "Relationship between Quality of Nursing Work Life and Uniformed Nurses’ Attitudes and Practices Related to COVID-19 in the Philippines: A Cross-Sectional Study"

_ijerph, 2021, doi:10.3390/ijerph18199953_

Round 1
Reviewer 1 Report
The following major issues must be addressed:
- English language used in the manuscript must be improved greatly. Many grammar and spelling mistakes throughout the manuscript.
- The Ethics Number for the study must be stated clearly in the manuscript.
- Table 1 must be revised. The mean of age is under a column for N and the S.D is under a column for %. This must be revised and redone clearly so readers can understand the Table well.
- A separate Methodology section must be put in the manuscript. Since the manuscript is a systematic review, the Methodology section is a very important part of the manuscript.
- Results obtained for the Philippines must be compared to other countries and regions in the Discussion section.
- Overall, the manuscript require major revisions before being considered for publication.
Author Response
1. The English language used in the manuscript must be improved greatly. Many grammar and spelling mistakes throughout the manuscript.
Thank you. The grammar and spelling has been checked, and the text has been amended. We hope that it is now better and clear.
2. The Ethics Number for the study must be stated clearly in the manuscript.
Thank you. Ethics number 0720-022 has been presented in ethical considerations.
3. Table 1 must be revised. The mean of age is under a column for N and the S.D is under a column for %. This must be revised and redone clearly so readers can understand the Table well.
Thank you. Table 1 was revised. The means and S.D of age was excluded. All the demographic features in Table 1 were represented as number (N) and percentage (%).
4. A separate Methodology section must be put in the manuscript. Since the manuscript is a systematic review, the Methodology section is a very important part of the manuscript.
Thank you. We revised the methodology with a separate section from 2.1 to 2.8.
5. Results obtained for the Philippines must be compared to other countries and regions in the Discussion section.
Thank you. We have compared the results with other countries such as Arabia, Jordan, India, Bangladesh, and Indonesia in our original manuscript. In the revision, we have further presented the information in introduction section and compared the country, China (this article was just published in July, 2021).
6. Overall, the manuscript requires major revisions before being considered for publication.
Thank you. We have made a major revision and hopefully the revision has been met the publication requirement.
Please see the attachment. Thank you.

Reviewer 2 Report
The Relationship between Quality of Nursing Work Life and Attitude and Practice towards COVID-19 among Philippine Uniformed Nurses: A Cross-Sectional Study
Comments and Suggestions for Authors
This study collected QNWL data of 147 nurses in four military hospitals in Manila during a period of half a year using two questionnaires. Results showed that these nurses had high QNWL and positive attitude in fighting COVID-19. To improve the work, the authors may consider following suggestions:
- Some basic quantitative information should be provided to justify the results in this article, including: how many hospitals are there in total, how many nurses in total in each hospital studied, how many COVID19 patients were taken care of by these four hospitals in total and each, and what is the percentage considering all COVID19 patients in this region.
- Since the infected cases and deaths toll have been changing during the research period of this article, a trend description of these statistical figures shall be presented. And also a stratified close look is necessary of each one of the four hospitals with further analysis and the patients situation, which would expose detailed information about the QNWL.
- Information presented in tables can be visualized with charts for better comparison and information transmission.
- As the authors discussed in section 4.1, income might influence QNWL. It is necessary to present segregate information concerning different interviewees’ marital status and family size, since more than 40% are married.
- Line 41-44, it seems that the two sentences “In a systematic review…….meta-analysis, respectively(7)” need further rephrasing.
- Line 45-48, what do the percentages mean exactly?
- Line 74, HWC?
- Line 155, the quality of figure 1 can be better.
- Line 205, the values of mean and SD in the table 3 may not be meaningful since the feedbacks to answers are not quantitative. There are similar situations in table 2 and table 4.
- Section 3.4, the three models need be presented in the second section to explain how they work.
- Section 4, the discussion on risk perception of COVID-19 of nurses is anticipated to be expanded in sub-sections.
- Section 4.1 the singles are 57% which may not be necessarily expressed as “the majority of … single”.
- Part of reference works can be moved to the introduction section to enrich background explanation.
Author Response
- Some basic quantitative information should be provided to justify the results in this article, including: how many hospitals are there in total, how many nurses in total in each hospital studied, how many COVID19 patients were taken care of by these four hospitals in total and each, and what is the percentage considering all COVID19 patients in this region.
Thank you for your reminder. There were four hospitals participating in the study. We did not provide some basic quantitative information, that’s because at that time we thought these variables were not associated. As data collection has already finished, we have presented our weakness in research limitations and suggest collecting relevant basic information in the future study.
- Since the infected cases and deaths toll have been changing during the research period of this article, a trend description of these statistical figures shall be presented. And also a stratified close look is necessary of each one of the four hospitals with further analysis and the patients’ situation, which would expose detailed information about the QNWL.
Thank you. As this is a cross-sectional study, it can only explore the phenomenon at a specific time during data collection. However, the trend can be predicted by statistical methods. To present the further analysis and outcomes of each hospital, the overall relationship between quality of nursing work life (QNWL) and attitudes and practices toward COVID-19 may be out of focus. In addition, due to page limitations, it may not be possible to fully present the further analysis results of each hospital. We will consider this suggestion as a reference for our next submission.
- Information presented in tables can be visualized with charts for better comparison and information transmission.
Thank you for your recommendation. Information presented in Table 1 and 2 was visualized with histogram and box plot.
- As the authors discussed in section 4.1, income might influence QNWL. It is necessary to present segregated information concerning different interviewees’ marital status and family size, since more than 40% are married.
Thank you. Different interviewees’ marital status and their family size has been discussed in page 46.
- Line 41-44, it seems that the two sentences “In a systematic review…….meta-analysis, respectively (7)” need further rephrasing.
Thank you. The sentence has been rephrased.
- Line 45-48, what do the percentages mean exactly?
Thank you. The percentage has been excluded from the manuscript.
- Line 74, HWC?
Thank you. It should be HCW instead.
- Line 155, the quality of figure 1 can be better.
Thank you. Figure 1 has been improved.
- Line 205, the values of mean and SD in table 3 may not be meaningful since the feedback to answers are not quantitative. There are similar situations in table 2 and table 4.
Thank you. We merge table 2 and 3 and remove table 3 respondent’s feedback. We have also removed mean and SD in table 4.
- Section 3.4, the three models need to be presented in the second section to explain how they work.
Thank you. The three models were represented and further explained in the second section.
- Section 4, the discussion on risk perception of COVID-19 of nurses is anticipated to be expanded in sub-sections.
Thank you. We have presented the risk perception of COVID-19 of nurse in sub-sections.
- Section 4.1 the singles are 57% which may not be necessarily expressed as “the majority of … single”.
Thank you. The expression has been modified by writing the actual quantifier “More than half”
- Part of reference works can be moved to the introduction section to enrich background explanation.
Thank you. We have moved some references to background section and made further explanations to enrich the importance of background.
Please see the attachment. Thank you.

Reviewer 3 Report
This paper was a study on the Philippines' countermeasures against COVID-19 for Filipino nurses.
(1) In the case of COVID-19, nurses' hard work is also an important factor, but one of the practical solutions is the presence or absence of vaccination. This study did not take into account nurses' opinions on vaccinated patients and vaccines for COVID-19, and this is an area that needs further improvement.
(2) The nurse's mind (QNWL) and professionalism are of course important as a countermeasure against COVID-19. Since it is a virus with so much mutation, it is considered as an important factor to develop a principled treatment or an effective vaccine. However, this study simply approaches the nurse's mind and has limitations in this area. I think it would be good to include this content in the limit part of the research, which is presented at the end of a typical research paper.
Author Response
- In the case of COVID-19, nurses' hard work is also an important factor, but one of the practical solutions is the presence or absence of vaccination. This study did not take into account nurses' opinions on vaccinated patients and vaccines for COVID-19, and this is an area that needs further improvement.
Thank you. Vaccination has been taken into account in the limitation of the research.
- The nurse's mind (QNWL) and professionalism are of course important as a countermeasure against COVID-19. Since it is a virus with so much mutation, it is considered as an important factor to develop a principled treatment or an effective vaccine. However, this study simply approaches the nurse's mind and has limitations in this area. I think it would be good to include this content in the limit part of the research, which is presented at the end of a typical research paper.
Thank you. The factor of vaccination has been mentioned in the limitation of the research.
Please see the attachment. Thank you.

Round 2
Reviewer 1 Report
The manuscript is ready for publicationReviewer 2 Report
The authors made meaningful revisions accordingly from the response letter. I believe the paper was improved.